**communications** engineering

# Triple-chord trussed submerged floating tunnels: hybrid construction concept, feasibility and design
Fa-Cheng Wang [1,2] ✉, Tao Zhuge [1], Zheng-Qing Cheng[1,3], Lin-Hai Han[1], Jian-Min Zhang[1,2] & Leroy Gardner [1,4] ✉

Submerged floating tunnels (SFT) offer a promising solution for deep-water crossings and intercontinental transportation. However, current SFT designs struggle to meet the high structural performance demands associated with the harsh service environments while remaining economically viable, thus limiting their implementation in practice. Here, we propose a conceptual SFT design using a triple-chord trussed concrete-filled double-skin tubular (CFDST) hybrid structure, featuring CFDST chords and hollow steel tubular braces. This design is highly adaptable and allows the steel tubes and sandwiched concrete to work synergistically, achieving efficiency in withstanding multiple loading conditions including lateral flow, internal fire, fatigue and impact loading. We further develop a multi-scale structural analysis methodology that integrates three-dimensional solid finite element (3-D FE) and simplified fibre modelling for the efficient evaluation of global deformations, fire performance and joint behaviour. The results demonstrate that the proposed design leads to considerably enhanced resistance against lateral flow loading, vibrations and internal fire, and is more adaptable and cost-effective than existing solutions.

As the economic connections between coastal regions strengthen, there has been an increasing demand for crossing deep or expansive water areas such as Norway's fjords, Japan's Funka Bay, Italy's Strait of Messina and China's Qiongzhou Strait. Constructing transportation infrastructure through such water areas can be challenging and expensive for traditional trans-water solutions such as cross-sea bridges and immersed tunnels. An alternative transportation option of using submerged floating tunnels (SFT) has recently emerged[1–3]. Such structures use self-buoyancy to neutralise their gravitational load, with mooring cables, similar to those used in pre-stressed structures[4], typically used to support the suspended tunnel body in the water. This system offers unique advantages over traditional trans-water infrastructure. For instance, the design of cross-sea bridges is primarily governed by the self-weight of the structure, with the majority of the material used to resist gravitational loads. This is not, however, the case in SFT design, thereby allowing for material savings and potential cost advantages. Furthermore, other advantages such as shorter lengths, steadier longitudinal slopes, less interruption to maritime transportation, greater environmental sustainability and more stable unit length costs have also been reported in previous studies[2,3,5]. However, SFTs are also exposed to a unique combination of loading from both normal service and hazardous scenarios due to their distinct service environments, typically including:

- High-velocity lateral flows in fjord-like terrains, which can induce large lateral displacements that threaten both the operational safety and long-term serviceability of the tunnel. Excessive deformation under lateral flow may interfere with internal traffic or even compromise structural stability. Design guidelines for long-span transportation infrastructure typically impose strict displacement limits to mitigate such risks[6]. For SFTs directly exposed to marine currents, lateral displacement thus represents a fundamental performance concern and should be treated as a primary serviceability criterion in SFT design.

- Wave-induced vibrations, which give rise to cumulative fatigue damage, especially in structural systems featuring multiple connections, such as trusses. Unlike immersed tunnels, which benefit from soil confinement, SFTs remain continuously subjected to dynamic excitation from waves[1,5,7,8] and internal traffic[9]. Previous research has shown that fatigue cracking in joint regions often governs the service life of marine truss structures[10,11]. Accordingly, fatigue life at critical connection locations is commonly selected as a key durability metric in the evaluation of SFT designs.

[1]School of Civil Engineering, Tsinghua University, Beijing, PR China. [2]Institute for Ocean Engineering, Tsinghua University, Beijing, PR China. [3]Railway Engineering Research Institute, China Academy of Railway Sciences Corporation Limited, Beijing, PR China. [4]Department of Civil and Environmental Engineering, Imperial College London, London, United Kingdom. ✉e-mail: wangfacheng@tsinghua.edu.cn; leroy.gardner@imperial.ac.uk

- Internal fires occurring within the enclosed underwater space, which pose severe threats to occupant safety and structural integrity. Compared to open-air bridges, SFTs have more confined ventilation conditions, and their direct contact with external water increases the consequences of structural failure due to thermal degradation. A breach of the tunnel wall during a fire could result in uncontrolled water ingress and rapid flooding. For this reason, fire resistance is commonly regarded as a critical requirement in tunnel design practice, especially for the SFT structures[12–15].

These harsh loading conditions introduce heightened design complexity and operational challenges compared to conventional cross-sea structures. While SFT offer the potential for superior performance, addressing these challenges requires design innovation and thorough comparative analyses to ensure safety and feasibility. Meanwhile, although the mentioned scenarios—lateral flow, wave-induced vibration, and internal fire—do not represent the full spectrum of loads that an SFT may be subjected to, they are among the most representative and critical. Therefore, this study focuses on these three scenarios as the principal basis for performance evaluation.

In the past decade, typical SFT designs feature a single-tube structure. Lee et al.[16] determined the key parameters for predicting the seismic behaviour of SFT with rectangular profiles. Kristoffersen et al.[17] compared the different behaviour of SFT with circular and rectangular sections in internal blast scenarios. Jin et al.[8] evaluated the factors controlling SFT design with circular sections under wave and earthquake loading scenarios. Kim and Kwak[18] promoted a 3-D equivalent static approach to determine the geometry of circular SFT structures under wave loading. Torres-Alves et al.[9] assessed the reliability of circular SFT under traffic loads. Chung et al. and Jeong and Kim[7,19] used numerical simulations to investigate the performance of circular SFT under wave actions. Kim et al.[20] provided insight into the behaviour of circular SFT under tsunami loading. These studies demonstrated the feasibility of SFT solutions, but single-tube SFT face limitations in balancing the material cost and structural safety in harsh service environments, as design options for improving their resistance are limited. Typically, engineers can only achieve performance gains by increasing the wall thickness, which may lead to prohibitive material costs. Overall, the simple structure of single-tube designs limits their adaptability to harsh service environments, hindering their practical application.

In response to these challenges, we propose a conceptual SFT design using a triple-chord trussed concrete-filled double-skin tubular (CFDST) hybrid structure. A schematic view and the geometric details of the proposed structure, which uses CFDST chords and hollow steel tubular braces, are shown in Fig. 1. The CFDST members consist of coaxial inner and outer steel tubes with concrete infilled between them. The key benefits of the proposed structural system relate to the synergistic effects between the concrete and steel, and can be summarised as follows:

- The sandwiched concrete can provide support to the outer steel tube, delaying the occurrence of local buckling[21–23], provide blast protection by absorbing the blast energy[24,25], provide heat insulation in the fire[12,13,26], preventing the fire damage extending and enhance the fatigue resistance of the joint regions in the truss structure[10,11,27].
- The outer steel tube can provide confinement to the sandwiched concrete, enhancing the concrete strength[28–30] and delaying the development of concrete cracking[31,32].

**Fig. 1 | Schematic view and design details for the proposed structure. a** Conceptual view of the typical design and proposed design of submerged floating tunnels. The typical design comprises a single-tube structure, while the proposed design comprises a trussed structure. The brown parts give the typical segments of the two designs. **b** Design details of the proposed structure. ①: concrete-filled double-skin tubular (CFDST) chord; ②: Horizontal brace; ③: Sloping brace. The CFDST chords are made of an inner steel tube, an outer steel tube and sandwiched concrete. The braces are made of hollow steel tubes. $L_i$: distance interval for each joint; $L_s$: length of the tunnel; $B$: distance between upper chords; $H$: section height; $d_i$: inner diameter of inner chord tube; $t_c$: wall thickness of sandwiched concrete; $t_o$: wall thickness of outer chord tube; $t_i$: wall thickness of inner chord tube; $d_s$: inner diameter of sloping brace tube; $t_s$: wall thickness of sloping brace tube; $d_h$: inner diameter of horizontal brace tube; $t_h$: wall thickness of horizontal brace tube.

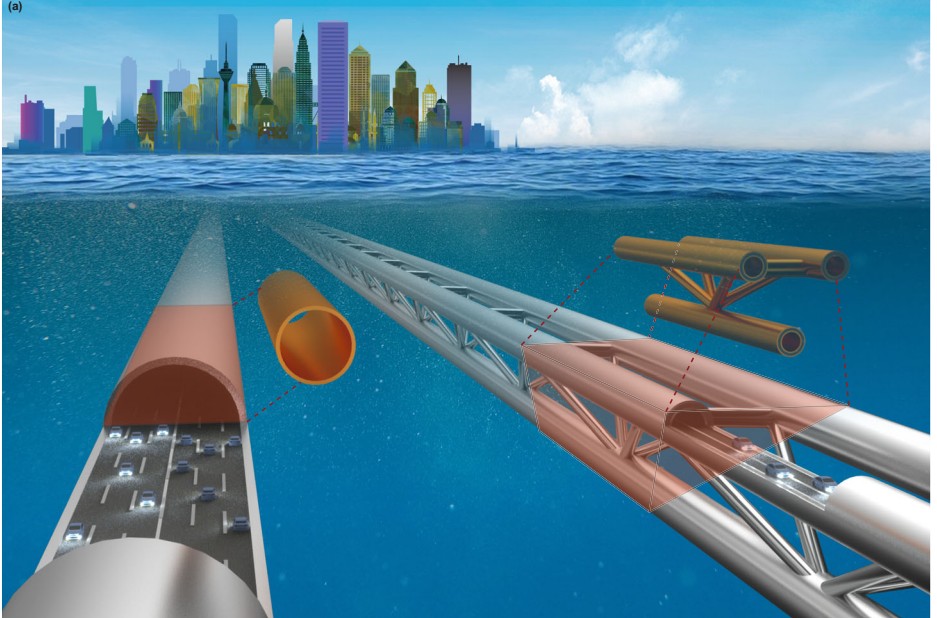

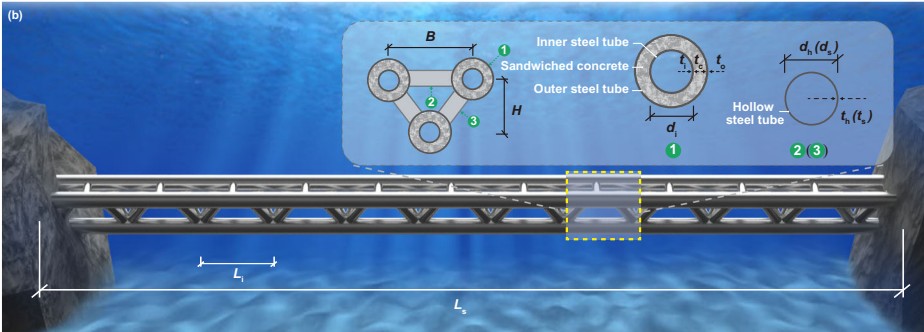

- The inner steel tube can provide support to the sandwiched concrete, developing effective confinement action in conjunction with the outer steel tube[33,34], and can provide protection against explosive spalling of the concrete[14,35] in fire.

To promote the implementation of the proposed SFT, it is essential to determine the design parameters based on the structural performance under harsh service conditions. Previous systematic studies on steel-concrete composite structures have been conducted, including investigations into the confinement effects[28–30], the steel-concrete interfacial behaviour[36–38], the fatigue performance in the joint regions[10,11,27] and temperature effects[12,13,26]. Building on these efforts, researchers have developed a structural analysis methodology based on three-dimensional solid finite element (3-D FE) analysis to evaluate the performance of isolated elements (such as columns and beams) under multiple loading conditions[39–42]. Typically, 3-D solid elements are used to simulate the concrete and shell elements are used to simulate the steel tubes[43]; this modelling approach has also been successfully applied to analyse the mechanical behaviour of numerous onshore structures[44], gaining broad recognition for its applicability[45,46]. However, when applied to long-span structures, such as the proposed SFT, the current analysis methodology faces the challenge of prohibitive computational costs.

To address this, we propose a multi-scale structural analysis methodology, where a simplified fibre modelling method based on beam elements is developed and integrated into the current analysis methodology to evaluate the deformation response of the SFT structure at the global scale.

Based on the described methodology, this paper assesses the performance and feasibility of the proposed SFT under the harsh loading conditions of lateral flow, vibration and internal fire scenarios. Key performance indicators, including the maximum lateral displacement, the fatigue life of the joint regions and the temperature distribution under internal fire loading, are analysed and compared against three conventional SFT designs. Furthermore, a parametric sensitivity study is conducted to assess the performance of the proposed SFT in terms of both the structural response and cost-effectiveness.

## Results
### Multi-scale structural analysis methodology
To ensure the safety of the proposed SFT structural system, it is essential to accurately and efficiently evaluate both its local performance (such as the fatigue life of the joint regions under vibration and the temperature distribution under internal fire) and its global performance (such as the global structural deformation) under various loading conditions. Therefore, we propose a multi-scale structural analysis methodology incorporating 3-D FE modelling and simplified fibre modelling to evaluate the structural performance across the local and global scales. In this paper, the numerical analysis is conducted through the finite element software ABAQUS[47], which has been successfully employed in similar previous studies[48].

**3-D FE modelling.** Wang and Han et al.[45,46,49–51], through extensive experimental and numerical studies, developed a systematic structural analysis methodology for steel-concrete composite structures, which accurately captures the confinement effects between the two materials. Concrete damaged plasticity (CDP) and secondary flow plasticity models are employed to simulate the plastic behaviour of the concrete and steel, respectively. The Coulomb friction model is applied to simulate the bond-slip behaviour at the steel-concrete interface. This refined modelling method is able to simulate the detailed structural response of steel-concrete composite structures and is suitable for small-scale structural analysis. In this paper, we use this method to evaluate the local structural performance of the studied SFT structural system, including the fatigue life of the joint regions under vibration and the temperature distribution under internal fire scenarios.

**Simplified fibre modelling.** To provide the necessary enhancements in computational efficiency to facilitate global analysis of the SFT structure,

it is necessary to utilise beam finite elements for the discretisation and to simplify the material and steel-concrete interface contact models. The following simplifying assumptions are made:
- The SFT remains linear elastic.
- The cross-sectional planes remain plane after structural deformation.
- The confinement afforded to the concrete by the steel tubes is conservatively ignored.

Simplified fibre models of the proposed SFT structure are developed using PIPE31H elements[47], with the CFDST chords transformed into single-material members by adjusting the inner and outer diameters of the beam elements following equivalence principles[52]. The validity of the simplified fibre modelling method is demonstrated through comparisons with the 3-D FE modelling method; the details are provided in the "Methods" section. It should be noted that this simplified modelling method primarily focuses on capturing the elastic displacement response of the SFT structures. However, the assumptions made in this simplification introduce limitations in accurately capturing detailed stress and strain distributions, which would require more advanced modelling methods, such as the 3-D FE modelling method.

### Numerical analysis results
Using the aforementioned multi-scale structural analysis methodology, we evaluate the proposed SFT in terms of:
- Maximum lateral displacement under lateral flow
- Fatigue life of the joint regions under vibration
- Temperature distributions under internal fire scenarios

We then design four SFT schemes for comparison purposes, namely:
- Scheme A: Single steel-only tubular structure
- Scheme B: Single CFDST structure
- Scheme C: Triple-chord steel-only tubular truss structure
- Scheme D (proposed in this paper): Triple-chord trussed CFDST hybrid structure

To ensure a fair comparison among the SFT schemes, the study assumes equal traffic flow bearing capacities and an equal material cost for each scheme, as shown in Fig. 2a. Specifically, each scheme is designed to withstand a 6-lane traffic flow and the quantities of concrete and steel for each design are determined based on a market price investigation reported by the Beijing Municipal Commission of Housing and Urban-rural Development in September 2024[53], where the price of concrete is 430 RMB m$^{-3}$ (60.38 USD m$^{-3}$) and the price of steel is 3803 RMB t$^{-1}$ (534.02 USD t$^{-1}$). This design criterion ensures that the overall material expenditure remains equivalent across the different schemes. It should be noted that this cost calculation considers only the raw material unit prices and does not incorporate additional factors such as fabrication effort, connection detailing or installation-related costs. The geometric details of the four schemes are listed in the "Methods" section. Additionally, we conduct a parametric sensitivity analysis on the maximum lateral displacement and cost under the lateral flow scenario to evaluate the adaptability and cost-effectiveness of the designs.

**Maximum lateral displacement under the lateral flow scenario.** To assess the performance of the proposed structure under lateral flow, we conducted a comparative analysis on the maximum lateral displacement between the four schemes, which are subjected to lateral fluid loads of the same flow velocity $v$. In this paper, we assume the SFT are to be constructed in the Strait of Messina; the flow velocity is therefore set to 3 (m s$^{-1}$)[54]. The fluid load is considered as a distributed load and is applied to the structures through ABAQUS/Aqua[47], as shown in Fig. 2b. Since no SFT structures have been constructed to date, we refer to the Chinese bridge code GB50917-2013[6] to further evaluate the structural feasibility. According to this standard, if the maximum displacement of a structure exceeds 1/1600 of its length, pre-camber should be incorporated in the design. Considering that this may increase construction costs and could

potentially lead to larger deformations when the lateral flow acts on the concave side of the camber, this study limits the maximum lateral displacement of the SFT to 1/1600 of its length $L_s$. This requirement means that the maximum lateral displacement must be limited to 625 mm.

Figure 2c, d shows the comparison of the maximum lateral displacement and the structural stiffness for all four schemes. This comparison is made under the condition that material costs and traffic flow bearing capacities are the same for all schemes. As such, the direct comparison of displacement and stiffness not only reflects the structural performance but also indirectly indicates the material efficiency of each design. Therefore, while normalising these parameters or comparing them against cost metrics is a valid approach in certain contexts, the direct comparison remains appropriate and informative for evaluating the designs in this study, as material cost equivalence ensures that any differences observed are due to the structural configurations themselves.

In this study, the stiffness $K_s$ is simply defined as the ratio of the support reaction force $R$ to the maximum lateral displacement $u_m$. It is found that the proposed structure can considerably reduce the maximum lateral displacement by about 60%, 35% and 25% relative to Scheme A,

Scheme B and Scheme C, respectively and enhance the structural stiffness by 400%, 200% and 35% relative to Scheme A, Scheme B and Scheme C respectively. Furthermore, as shown in Fig. 2c, the maximum lateral displacement of Schemes A and B exceeds the allowable limit, while Schemes C and D remain within the threshold. Scheme D comfortably satisfies the maximum lateral displacement requirements and could do so at a lower material cost.

**Fatigue life of joint regions.** Fatigue problems in the joint regions, commonly arising from vibrations induced by wave and internal traffic loads, pose a potential threat to the structural safety of the proposed design. To assess the fatigue performance in the joint regions of the proposed SFT design, we conduct a comparative study between the proposed design and the triple-chord steel-only truss structure. The latter is designed with the same material cost and outer chord diameters as the proposed SFT design. Both structures are subjected to a four-point bending configuration. The lower bounds of the cyclic loads are set to be 0 kN, while the upper bounds are set to vary from 30 kN to 70 kN, as shown in Table 1. The hotspot stress method is used to calculate the fatigue life[55].

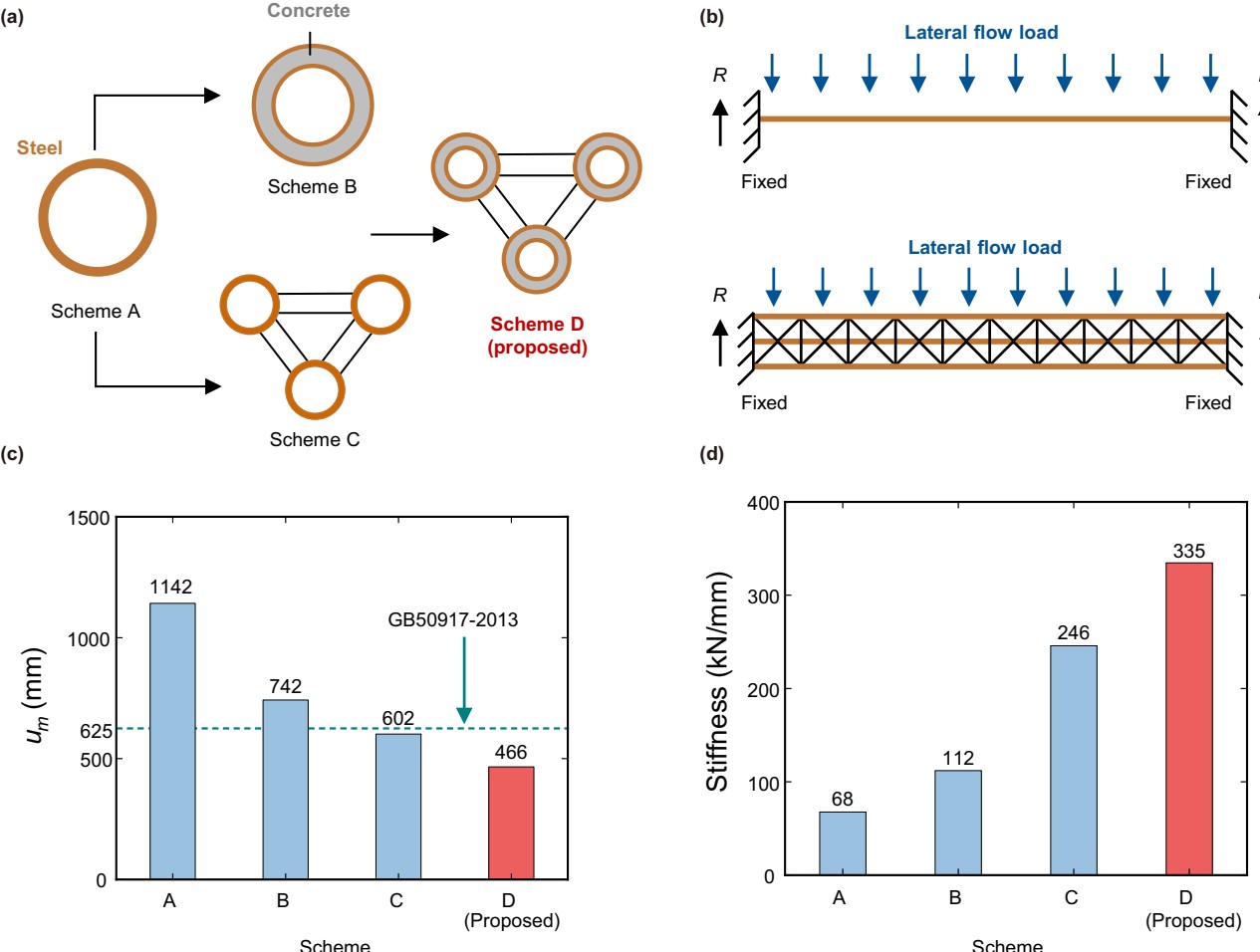

**Fig. 2 | Comparisons of maximum lateral displacement and stiffness under lateral flow load. a** Schematic view of the four submerged floating tunnel (SFT) schemes. Scheme A: single steel-only tubular structure; Scheme B: single concrete-filled double-skin tubular (CFDST) structure; Scheme C: triple-chord steel-only tubular truss structure; Scheme D: triple-chord trussed CFDST hybrid structure. The brown parts represent those made of steel, and the grey parts represent those made of concrete. The concept of the proposed SFT is to enhance the performance of the steel-only structure by using the combination of the CFDST tubes and truss structure. **b** Typical top view of the loading conditions to which the structures were subjected. The four schemes have all 6 degrees of freedom restrained (fixed support) at the two ends, and the lateral flow is treated as a distributed load and applied to the whole structure (blue arrows in the figure). The lateral reaction force at each end of the structure is defined as $R$ (black arrows in the figure). **c** Comparison of the displacement results for the four schemes. Red column represents the maximum lateral displacement ($u_m$) of proposed scheme, and blue ones represent that of the conventional designs. The threshold value of 625 mm is determined according to GB50917-2013[6] (green dashed line in the figure). **d** Comparison of the stiffness of the four schemes. Red column represents the stiffness of proposed scheme, and blue ones represent that of the conventional designs. The detailed calculation results can be found in Supplementary Table 1. The stiffness is calculated by $R/u_m$.

**Table 1 | Fatigue life calculation results**

| Structure type | Cyclic load range (kN) | Hotspot stress (MPa) | Fatigue life (logarithm value) | Fatigue life (10⁶ load cycles) |
|---|---|---|---|---|
| Triple-chord steel-only truss | 30 | 56.8 | 8.4 | 266.2 |
| | 40 | 75.8 | 7.8 | 62.9 |
| | 50 | 94.7 | 7.3 | 20.7 |
| | 60 | 113.7 | 6.9 | 8.3 |
| | 70 | 132.6 | 6.6 | 3.9 |
| Proposed | 30 | 59.2 | 8.7 | 525.4 |
| | 40 | 79.0 | 8.1 | 124.2 |
| | 50 | 98.9 | 7.6 | 40.4 |
| | 60 | 118.9 | 7.2 | 16.1 |
| | 70 | 138.8 | 6.9 | 7.4 |

The fatigue life is calculated based on the hotspot stress method and the S-N curve recommended by CIDECT[55].

A required fatigue life of 100 years[56] and a constant vibration period of 7s[57] are assumed.

Figure 3a shows the typical position of the hotspot stress within the sloping braces adjacent to the joint regions. Figure 3b represents the load-fatigue life relation of the proposed structures and triple-chord steel-only truss structures. Under the same cyclic load range, the proposed structures can considerably increase the fatigue life. To be more specific, the presence of sandwiched concrete helps to increase the fatigue life of the joint regions by approximately 94%.

Figure 3b also shows the feasibility check results. The proposed design meets the 100-year fatigue life requirement under cyclic load range of 30 kN, while the steel-only structure fails to do so. This suggests that the proposed SFT design can meet the structural fatigue life requirements under given fatigue loads at lower material costs compared to the conventional designs.

**Temperature distribution under the internal fire scenario.** In determining the temperature distribution under the internal fire scenario, it is assumed that the distance between the chords is sufficiently large that a fire within one chord will not influence the other two chords. In this case, the assessment of the fire performance of the proposed structure can be transformed into the assessment of the fire performance of CFDST members under internal fire. Hence, based on Scheme A and Scheme B, we compared the temperature field of the two schemes under the internal fire scenario. The RABT fire curve[15] is applied to the internal space of the structure.

Figure 4 shows the temperature distribution and typical temperature-time curves at representative positions through the wall thickness of the structure after a 1.5-h internal fire exposure duration. A considerable temperature rise can be observed on the inner side of both Scheme A and Scheme B. The maximum temperature on the inner surface of Scheme B is about 6% higher than that of the inner surface of Scheme A. However, most regions in Scheme B are barely affected by the internal fire, while a considerable temperature rise can be observed throughout the whole cross-section of Scheme A.

Considering that the SFT structure is loaded primarily in bending and that the outer steel tube generally contributes most to the flexural capacity and stiffness of the composite chord[41,58,59], we can reasonably infer from the comparison results that the mechanical performance of the proposed SFT will not severely degrade after 1.5 h of internal fire exposure, whereas the steel-only structure, if unprotected, will not be able to sustain the applied loading. Therefore, from the perspective of feasibility, achieving specific fire resistance requirements with a steel-only structure will necessitate higher material usage or the application of fire-resistant coatings, with considerable

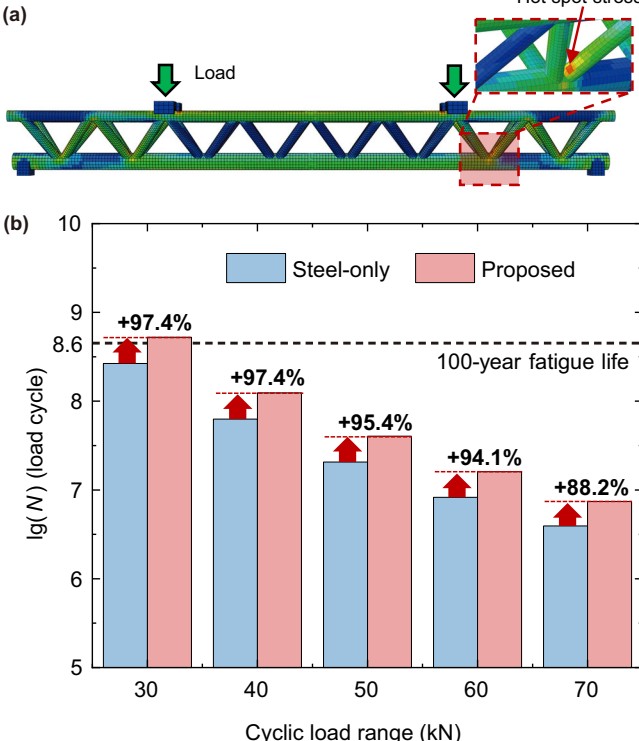

**Fig. 3 | Comparisons of fatigue life. a** Typical position of hotspot stress. The red dashed box shows the local enlarged image of the joint regions, and the red regions in one brace show the high-stress area in the joint regions, which is the typical location of hotspot stress. **b** Fatigue life under different cyclic load ranges and 100-year fatigue life threshold. The red columns show the fatigue life of the proposed scheme under different cyclic load ranges, and the blue ones represent that of steel-only designs. The black dashed line represents the 100-year fatigue life requirement. The percentage values shown above each pair of columns represent the calculated enhancement in fatigue life of the proposed design relative to the steel-only design for each load range. The fatigue life $N$ is transformed into its common logarithm value $\lg(N)$. The increase in fatigue life is determined based on the calculated $N$ value.

cost implications. In contrast, the proposed SFT benefits from the protective role of the sandwiched concrete to ensure structural fire safety.

**Parameter sensitivity analysis results.** To further illustrate the advantages of the proposed SFT in terms of adaptability and cost-effectiveness, we investigated the scenario where the lateral deformation of the Scheme D SFT structure was required to be reduced under the lateral flow condition. The lateral flow velocity is maintained at 3 m s⁻¹. In this case, we conducted a systematic parameter sensitivity analysis to illustrate the influence of the key design parameters on the proposed maximum lateral displacement and material cost of the structure. We have studied 13 design parameters; the results are summarised in Fig. 5. It should be noted that the maximum lateral displacement for all SFT designs in this section remains within the limit of $1/1600L_s$.

The results reveal that the ratio of maximum lateral displacement to the structural length $u_m/L_s$ is positively related to $L_s$, $H$ and $L_i$, and negatively related to the remaining 10 parameters, while the material cost to structural length $c_s/L_s$ ratio is negatively related to $L_i$ and positively related to the remaining 10 parameters. Among all the parameters mentioned above, three parameters—the length of the tunnel $L_s$, section height $H$, and distance between upper chords $B$—exert the most considerable influence on the maximum lateral displacement, while introducing negligible increases in material cost. These parameters are therefore recommended as primary variables for displacement control in practical design optimisation. Specifically, increasing $L_s$ and $H$ leads to higher displacement, whereas increasing

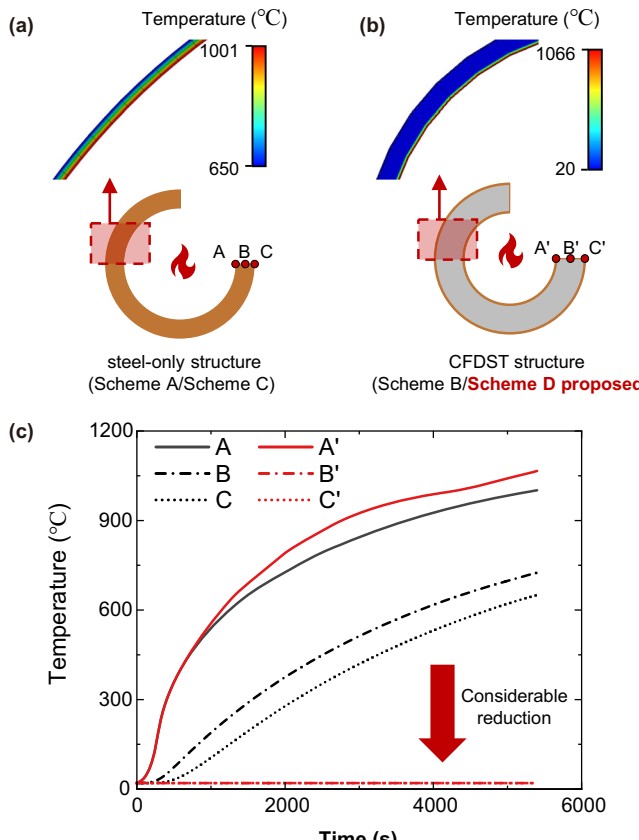

**Fig. 4 | Comparisons of temperature distribution under internal fire.**
**a** Temperature distribution for steel-only structure under internal fire. Point A is on the inner surface of the tube, point B is at the mid-thickness and point C is on the outer surface of the tube. The red dashed boxes indicate the temperature distributions in the structural sections. The colour in the section changes according to section temperature: from the minimum temperature to the maximum temperature of the section, the colour changes from blue to red in a linear pattern. **b** Temperature distribution for concrete-filled double-skin tubular (CFDST) structures under internal fire. The colour usage is the same as (**a**). **c** Typical temperature-time curves in representative positions of the section of the steel-only structure and CFDST structure. The red lines represent the temperature of the CFDST structure, and the black ones represent that of steel-only designs. The solid lines represent the temperature in point A/A', the dot dash lines show that in point B/B' and the dotted lines show that in point C/C'. The original simulation data can be found in Supplementary Data 1. Considerable reductions in temperature can be observed at points B' and C' of the CFDST structure.

*B* can mitigate this effect. Additionally, benefiting from the relatively lower price of concrete, increasing the thickness of the sandwiched concrete $t_c$ causes a limited cost rise but brings about a relatively notable increase in stiffness. Accordingly, $t_c$ may also be considered a favourable design parameter for improving lateral displacement performance. For the remaining nine parameters, given their low sensitivity and limited cost-benefit potential, design adjustments are not recommended for displacement control.

## Discussion

This study demonstrates that the proposed triple-chord trussed CFDST hybrid SFT design offers considerable structural performance advantages under typical and potential service scenarios, including lateral flow, vibration and internal fire, while maintaining material cost parity with existing solutions. These performance gains are primarily attributed to the synergistic action of the steel and concrete and to the truss-based configuration of the system.

From a structural perspective, the proposed design considerably enhances lateral stiffness and reduces deformation without excessive material usage. In particular, its response under lateral flow benefits from the flexibility of the truss layout and the efficient use of sandwiched concrete, which outperforms both single-tube and steel-only truss schemes. Furthermore, under cyclic loading, the integration of concrete in the joints mitigates stress concentrations and prolongs fatigue life—a conclusion consistent with previous findings on concrete-filled tubular joints[10,11,27]. In fire scenarios, the low thermal conductivity of the concrete acts as a natural insulator, protecting the outer tubes, which are critical for structural strength. This inherent fire resistance may allow designers to avoid expensive protective coatings required for steel-only alternatives.

Although increasing attention has been given to the structural performance of submerged floating tunnels in recent years, most existing studies remain focused on single-tube configurations[7–9,16–20]. Research on steel-concrete composite schemes is still limited[15], and investigations of trussed hybrid systems are even rarer. However, single-tube solutions face severe challenges in harsh marine environments, and no SFT projects have yet been successfully realised in practice. Our design exemplifies the potential of hybrid composite systems in addressing the multifaceted demands of subsea infrastructure. The study provides proof of concept for combining structural and economic efficiency through composite interaction and a rational geometric configuration, offering a promising design option for the practical application of SFT systems. Additionally, the use of a multi-scale numerical analysis methodology contributes a feasible framework for evaluating such complex systems that balances computational efficiency with modelling depth.

Several limitations of this study must also be acknowledged. First, while the simplified fibre modelling method offers computational efficiency for global analysis, it cannot capture local stress/strain distributions with sufficient precision, which would require full 3-D finite element modelling. Second, the parametric sensitivity analysis is conducted in an OFAT (One-Factor-at-a-Time) manner, which may neglect potential interaction effects among parameters. Future work should adopt design-of-experiments (DOE) or multi-variable optimisation techniques to provide a more holistic understanding of the parameter space. Third, the present comparison assumes equal material costs based on unit prices, without considering the differences in construction complexity, fabrication methods or connection detailing, which may influence total costs in practice. A more comprehensive economic analysis incorporating these factors is recommended to be pursued in future work. Fourth, while full-scale geometry is adopted for the lateral displacement, thermal, and parameter sensitivity analyses, a scaled-down model is used in the fatigue analysis. This is because accurate fatigue assessment requires capturing hotspot stresses, which simplified fibre models cannot provide, and full-scale fatigue simulations are extremely time-consuming. It is acknowledged that fatigue performance generally degrades with increasing scale due to material imperfections and enlargement of high-stress zones. As such, the small-scale model may yield an overestimated fatigue life. However, since this part of the study focuses more on comparative trends across schemes, the adopted scale remains suitable for relative performance evaluation. A systematic scale-effect analysis—integrating theoretical modelling, physical testing and numerical simulation—would provide valuable insights and is recommended for future work. Finally, while the proposed design demonstrates better performance in the numerical evaluations, several implementation challenges merit attention. The composite double-skin configuration—particularly the use of CFDST tubes with truss bracing—may introduce complexities compared to steel-only design in prefabrication, construction and underwater assembly. Ensuring reliable concrete filling, joint integrity and corrosion protection under deep-sea conditions presents non-trivial engineering challenges. Additionally, long-term maintenance and durability issues, especially in aggressive marine environments, could influence lifecycle performance. These aspects lie beyond the scope of the current study but are critical for future experimental validation and engineering applications.

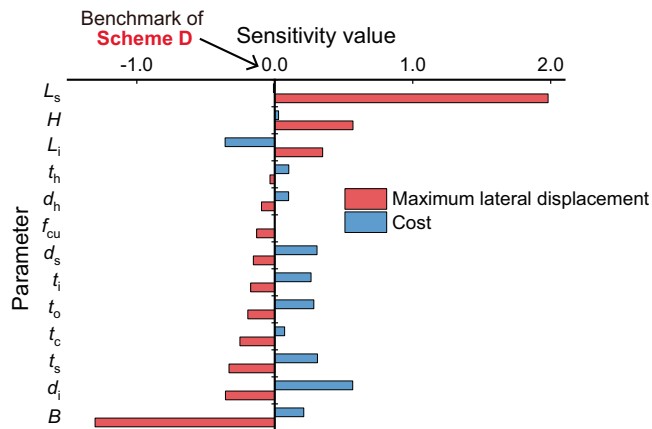

**Fig. 5 | Parameter sensitivity value for maximum lateral displacement and cost of the proposed structure.** The analysis benchmark is the Scheme D design (at origin point); The red bars represent the maximum lateral displacement sensitivity value of different parameters of the proposed design, and the blue ones represent the cost sensitivity value of different parameters of the proposed design. The definition of each parameter should be referred to Fig. 1. Table 2 gives the detailed design parameters, while the detailed simulation results can be found in Supplementary Data 2. The sensitivity value is defined as the ratio of the relative variance in the maximum lateral displacement ($u_m$) and cost to the relative variance in the studied parameters; further information can be found in the "Methods" section.

## Methods

The methodology adopted in this study involves a multi-scale modelling framework that integrates different numerical approaches to evaluate the performance of the proposed hybrid SFT structure under distinct loading conditions. As summarised in Fig. 6, we focus on three critical service scenarios—lateral flow, wave-induced vibration, and internal fire—each of which utilises a separate to a simulation approach. Specifically, 3-D FE modelling is used to analyse the local structural behaviour, including fatigue life in joint regions and temperature distribution under fire. In parallel, a simplified fibre modelling method is employed to evaluate the global elastic response of the full SFT structure, particularly the lateral displacement under hydrodynamic loading.

### Parameter determination for 3-D FE modelling

The concrete damage plasticity (CDP) model and the secondary flow plasticity model are adopted[47] to simulate the mechanical behaviour of the concrete and steel, respectively. This simulation approach considers the confinement effects in the concrete, offering convenience in modelling and high computational accuracy[49]. The thermal properties of the concrete and steel are determined according to T.T. Lie's recommendations[60], while the film coefficient and emissivity are determined according to Eurocode 4[61].

The CDP model parameters are determined following the recommendations set out by Han et al.[37]: the value of the dilation angle ($\psi$), the flow potential eccentricity ($e$), the ratio of the compressive strength under the biaxial loading to uniaxial compressive strength ($f_{b0}/f_{c0}$) and the ratio of the second stress invariant on the tensile meridian to that on the compressive meridian ($K$) are set as 30, 0.1, 1.16, 2/3 respectively. The provisions of ACI 318[62] are utilised to determine the modulus of elasticity $E_c$ and Poisson's ratio $\mu_c$ of the concrete, with $E_c = 4730(f_c'^{0.5})$ and $\mu_c = 0.2$.

The material behaviour of the high-strength steel is described using the bilinear model introduced by Li et al.[59]. This material model uses $0.01E_s$ as the modulus for the strain-hardening stage, which provides good accuracy, though yet greater accuracy could be achieved following the recent model proposed by Dissanayake et al.[63]. The modulus of elasticity $E_s$ and Poisson's ratio $\mu_s$ of the steel are determined according to GB50017-2017[64], with $E_s = 206$ GPa and $\mu_s = 0.3$.

Four-noded shell elements with reduced integration (S4R) and eight-noded solid elements with reduced integration (C3D8R) are employed to simulate the steel tubes and sandwiched concrete, respectively. A surface-to-surface interaction model is established to simulate the contact behaviour between the different components of the chord tubes. Specifically, the outer surfaces of the inner tubes and the inner surfaces of the outer tubes are set as master surfaces, while the inner surfaces and outer surfaces of the sandwiched concrete are set as slave surfaces. For each surface-to-surface contact pair, both normal and tangential behaviour are considered. The normal behaviour is modelled with "Hard" contact, which prohibits penetration between the surfaces in their normal directions. The tangential behaviour is modelled with the Coulomb friction model[37].

The edges of the hollow steel braces are connected to the outer surfaces of the chord tubes with the *Tie command to simulate rigid joints. The *Tie command is also used to connect the end plates, supporting blocks and loading blocks to the chord tubes.

### Parameter determination for simplified fibre modelling

In this study, we assume the axial and flexural stiffnesses of the chord section are invariant and allow the inner and outer diameters of the equivalent pipe section to be adjusted accordingly. The expressions for such transformations are as follows:

$$E_t A_t = E_o A_o + E_i A_i + E_c A_c \tag{1}$$

$$E_t I_t = E_o I_o + E_i I_i + E_c I_c \tag{2}$$

where $E_t$, $E_o$, $E_i$ and $E_c$ are the moduli of elasticity for the equivalent section, outer steel tube, inner steel tube and sandwiched concrete, respectively; $A_t$, $A_o$, $A_i$ and $A_c$ are the cross-sectional areas for the equivalent section, outer steel tube, inner steel tube and sandwiched concrete respectively; $I_t$, $I_o$, $I_i$ and $I_c$ are the second moments of area for the equivalent section, outer steel tube, inner steel tube and sandwiched concrete respectively.

Assuming the equivalent section is also made of steel, the outer diameter $D_{eq,o}$ and inner diameter $D_{eq,i}$ can be calculated as:

$$D_{eq,o} = \sqrt{\frac{I + A^2}{2A}} \tag{3}$$

$$D_{eq,i} = \sqrt{\frac{I - A^2}{2A}} \tag{4}$$

where $A = (4/\pi)(A_o + A_i + \alpha_c A_c)$, $I = (64/\pi)(I_o + I_i + \alpha_c I_c)$ and $\alpha_c = E_c/E_s$. The *Tie method is also used to connect the braces and chords.

### Validation of 3-D FE modelling method

Owing to the lack of physical experimental data on the proposed SFT structures, we used experimental results on similar structures to validate the 3-D FE modelling method. The validation process involved simulating the following two groups of specimens:

Group 1: Four-point bending tests conducted by Li et al.[59] on CFDST beam specimens with circular high-strength steel outer tubes and circular normal-strength steel inner tubes. The established model is simply supported at two supporting blocks, and the load is added to the specimens through an imposed vertical displacement, as illustrated in Fig. 7a.

Group 2: Four-point bending tests conducted by Han et al.[65] on two triple-chord trussed concrete-filled steel tubular hybrid structures (CFST trusses). The CFST chords and the hollow circular braces are constructed from normal-strength steel. The boundary conditions for the established model are similar to those for the models employed in Group 1. Figure 7c shows an overview of the numerical model.

The material properties and ultimate moment capacities of all specimens are summarised in Supplementary Table 2. The ultimate moments $M_{u,test}$ are defined as the cross-sectional moments at the mid-span of the specimens when the maximum strain in the bottom chord reaches 0.01[59,65].

The validation results of the 3-D FE modelling method are illustrated in Fig. 7. The global and local failure modes predicted by the 3-D FE models are

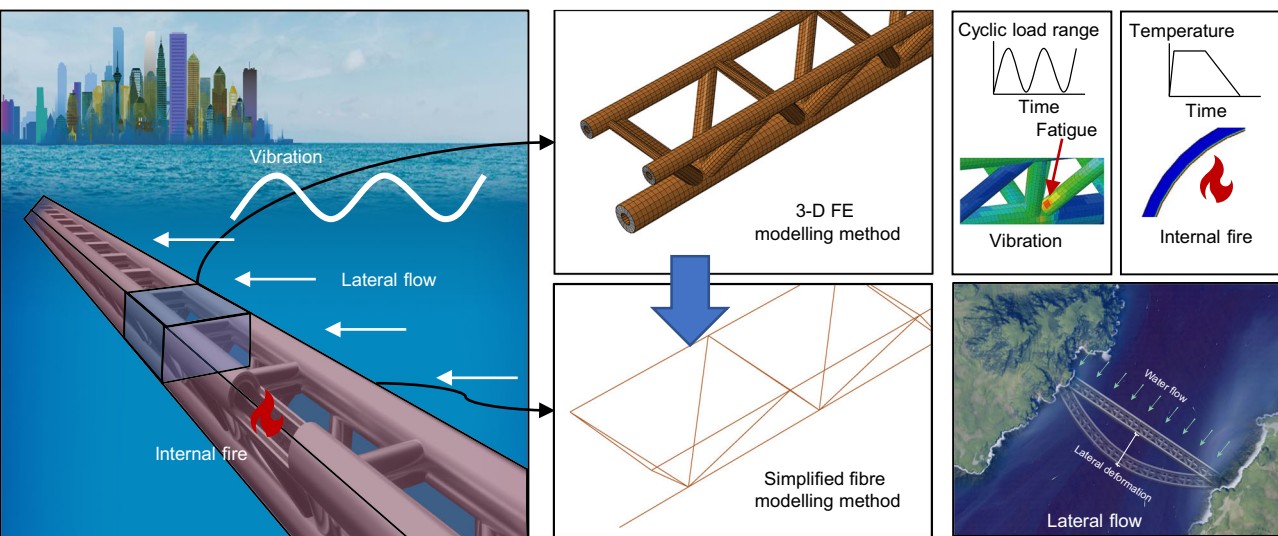

**Fig. 6 | Summary of the methodology of the research.** Unique loading combinations of lateral flow, vibration and internal fire are investigated. We use three-dimensional solid finite element (3-D FE) modelling to simulate the local behaviour of the structure, including the fatigue life of the joint regions and the temperature distribution under internal fire loading, and simplified fibre modelling to simulate the global behaviour, including determining the maximum lateral displacement.

in good accordance with those from the tests for both the CFDST beam specimens and the CFST truss specimens. The full-range load-deformation curves are compared in Fig. 7d. The initial stiffness, ultimate bending moment and the hardening stage of the numerical curves match well with the test results. Supplementary Table 2 summarises the ultimate bending moments predicted by the 3-D FE models $M_{u,pred}$ and those obtained from the experiments $M_{u,test}$. The mean value of $M_{u,pred}/M_{u,test}$ is 1.008, and the coefficient of variation (COV) is 0.035. The results show that the proposed 3-D FE modelling method can accurately predict the observed structural behaviour in terms of failure mode, full-range load-deformation response and ultimate load-carrying capacity, making it a viable reference for validation of the simplified fibre modelling method.

**Validation of simplified fibre modelling method**
Based on the validated CFST truss model, we designed five prototypes of the proposed structure with different slenderness ratios $\lambda$, which is modified by changing the length of the structure $L_s$ from 4.8 m to 14.4 m, as illustrated in Supplementary Table 3. We then developed two sets of numerical models for the five prototypes using 3-D FE and simplified fibre modelling methods. The deformation results from both sets are compared to complete the validation. The 3-D FE models are established by replacing the CFST chord tubes with CFDST members. The hollow ratio $\chi = (d_i + 2t_i)/(d_i + 2t_i + 2t_c)$ for the chords is set to 0.49, matching that of the test specimen B2. Each model is subjected to the same vertical load of 500 kN, and no material plasticity is considered.

To validate the proposed simplified fibre modelling method, we compared the structural deformation predictions from the 3-D FE model with those from the simplified fibre model, as illustrated in Fig. 8. Both models captured the overall bending behaviour. Figure 8b summarises the maximum displacement results from the 3-D FE models and simplified models. It is found that as the slenderness ratio increases, the relative error $\Delta$ decreases rapidly. When the slenderness ratio exceeds 50, the relative error is less than 3%. Considering that the proposed SFT structures are usually designed with a large slenderness ratio, the simplified fibre modelling method can accurately predict the maximum displacement of such structures. The above comparisons suggest that the simplified model is validated in terms of global deformation patterns and maximum displacement of the structure, making it a suitable substitute for the 3-D FE model for bending deformation predictions. Therefore, this study applies such a modelling method to investigate the structural deformation behaviour.

**Parameter design of SFT schemes**
Assuming the SFT schemes can withstand a 6-lane traffic flow and share the same material cost, we designed four SFT schemes based on the Chinese standard JTG 3370.1-2018[66] and the material price investigation reported by the Beijing Municipal Commission of Housing and Urban-rural Development in September 2024[53]. Specifically, we calculate the material costs with the following equations:

For single-tube schemes:

$$\begin{cases} P_s = \frac{\pi}{4}\gamma_s\rho_s L_s\left[(d_i+2t_i)^2+(d_i+2t_i+2t_c+2t_o)^2-d_i^2-(d_i+2t_i+2t_c)^2\right] \\ P_c = \frac{\pi}{4}\gamma_c L_s\left[(d_i+2t_i+2t_c)^2-(d_i+2t_i)^2\right] \\ P = P_s + P_c \end{cases}$$
(5)

For triple-chord trussed schemes:

$$\begin{cases} P_s = \frac{3\pi}{4}\gamma_s\rho_s L_s\left[(d_i+2t_i)^2+(d_i+2t_i+2t_c+2t_o)^2-d_i^2-(d_i+2t_i+2t_c)^2\right] \\ P_c = \frac{3\pi}{4}\gamma_c L_s\left[(d_i+2t_i+2t_c)^2-(d_i+2t_i)^2\right] \\ P_b = \frac{\pi}{4}\gamma_s\rho_s\{n_s L_{sb}[(d_s+2t_s)^2-d_s^2]+n_h B[(d_h+2t_h)^2-d_h^2]\} \\ L_{sb} = \sqrt{(L_i/2)^2+(B/2)^2+H^2} \\ P = P_s + P_c + P_b \end{cases}$$
(6)

where, $P$ represents the material cost of the scheme; $P_s$, $P_c$ and $P_b$ denote the material price of the steel used in chord tubes, the concrete used in chord tubes and the steel used in the braces respectively; $\gamma_s$ and $\gamma_c$ represent the unit price of steel and concrete respectively; $n_s$ and $n_h$ denote the number of sloping braces and horizontal braces used in the truss structures respectively; $L_{sb}$ is the length of a single sloping brace.

The geometric details are listed in Table 2. The four schemes are placed 50 m below the water surface. Hence, the flow load will be dominant, and the wave load will be small enough to be omitted[1]. To simplify the analysis, only the lateral uniform flow load is considered in this study. The flow load is applied to the structure through ABAQUS/Aqua[47]. The drag coefficient ($C_d$) is set to 1.2[67]. According to the recommendations from the Chinese standard JTS/T 144-1-2010[68], we use the maximum flow velocity in the construction site for the flow load calculation. Assuming the structure is constructed in the Strait of Messina, the lateral flow velocity ($v$) is set to 3 (m s$^{-1}$)[54]. The

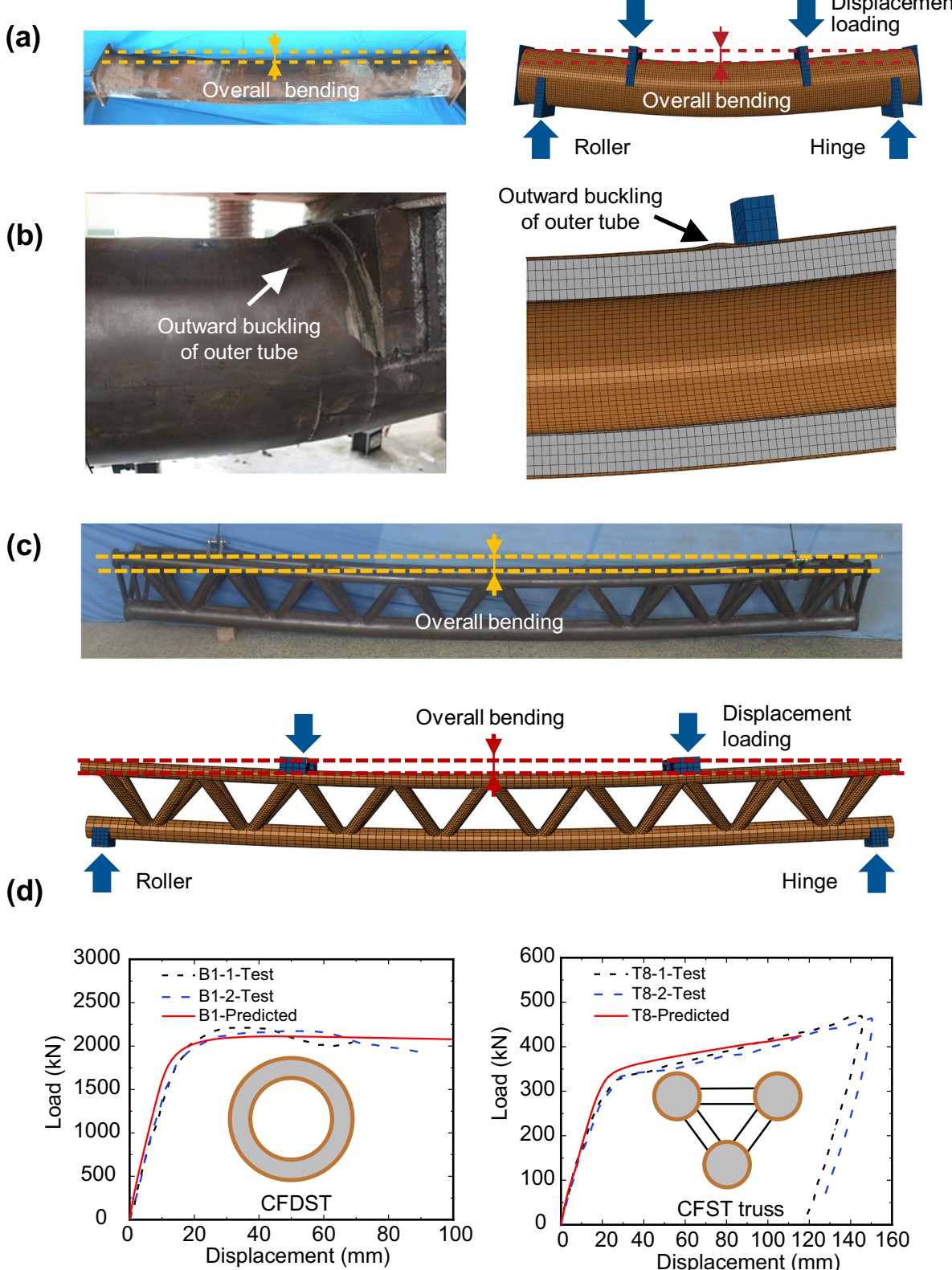

**Fig. 7 | Validation of three-dimensional solid finite element (3-D FE) modelling method. a** Comparison between test and 3-D FE global failure mode of typical concrete-filled double-skin tubular (CFDST) specimens (specimen B1). The models are supported on blocks: one has all translational degrees of freedom restrained (a hinge support), while the other has only its vertical displacement restrained (a roller support). The experimental figure originates from Li et al.[59]. **b** Comparison between test and 3-D FE local failure mode of typical CFDST specimen. The experimental figure originates from Li et al.[59]. **c** Comparison between test and 3-D FE failure modes of triple-chord trussed concrete-filled steel tubular (CFST) hybrid structure (specimen T8). The model is supported by one roller support and a hinge support similar to (**a**). The experimental figure originates from Han et al.[65]. **d** Comparison of typical full-range load-deformation curves. The load is defined as the reaction force obtained from the loading blocks; the displacement is defined as the displacement of the bottom point at the mid-span of the specimen. The black dot dash lines and blue long dash lines represent the two parallel test results collected from Li et al.[59] and Han et al.[65]. The red solid lines represent the 3-D FE model predictions of the specimens.

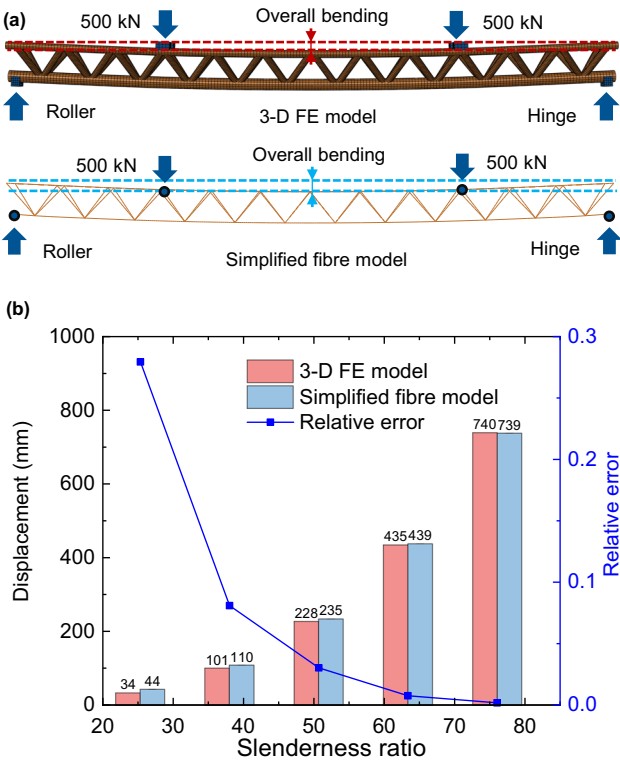

**Fig. 8 | Validation of simplified fibre modelling method. a** Comparison of the typical global deformation patterns. Both models are supported by end blocks or supporting points: one has all translational degrees of freedom restrained (a hinge support), while the other has only vertical displacements restrained (a roller support). The models are subjected to the same vertical load of 500 kN, and their deformation patterns and maximum displacement values are compared. **b** Comparison of maximum displacement. The red columns represent the displacement results predicted by the three-dimensional solid finite element (3-D FE) models ($u_{m,\text{3-D FE}}$), and the blue columns represent the displacement results predicted by the simplified fibre models ($u_{m,\text{simplified}}$). The solid blue line represents the relative error between simplified fibre models and 3-D FE models. The relative error is defined as: $|u_{m,\text{simplified}} - u_{m,\text{3-D FE}}|/u_{m,\text{3-D FE}}$.

structure is restrained in all degrees of freedom at both ends, with no other supports applied along the span. For all schemes, the yield strength of the steel $f_y$ and compressive cube strength of the concrete $f_{cu}$ are set to 345 MPa and 30 MPa, respectively.

## Acquisition of the maximum lateral displacement
We use the validated simplified fibre modelling method to establish the four SFT schemes listed in Table 2. The four structures are fixed at both ends and subjected to the lateral flow load. The maximum lateral displacement is acquired from the simulation results.

## Acquisition of the fatigue life of joint regions
We use the 3-D FE model of L-1 for comparison (Supplementary Table 3 gives the geometric details). The fatigue life of the joint regions is calculated using the hotspot stress method[55]:

We first subjected the structure to four-point bending, as shown in Fig. 3a, and extracted the maximum stress in the joint regions as the hotspot stress. Then, based on the hotspot stress, we used the S-N curve recommended by CIDECT[55] to calculate the fatigue life of the joint regions.

## Acquisition of the temperature distribution
We use the 3-D FE model of Scheme A and Scheme B to assess the structural performance under the internal fire scenario. For analysis convenience, we assume the fire is applied throughout the whole inner space of the structure;

### Table 2 | Geometric parameters for SFT schemes

| Parameters | | Value (unit) |
|---|---|---|
| **Single steel-only tube structure (Scheme A)** | | |
| Outer diameter of outer tube $d_o$ | | 27.92 (m) |
| Wall thickness of outer tube $t_{co}$ | | 209 (mm) |
| Length of the tunnel $L_s$ | | 1000 (m) |
| **Single CFDST structure (Scheme B)** | | |
| Outer diameter of outer tube $d_o$ | | 30.07 (m) |
| Inner diameter of inner tube $d_i$ | | 27.50 (m) |
| Wall thickness of outer tube $t_o$ | | 92 (mm) |
| Wall thickness of inner tube $t_i$ | | 92 (mm) |
| Wall thickness of sandwiched concrete $t_c$ | | 1100 (mm) |
| Length of the tunnel $L_s$ | | 1000 (m) |
| **Triple-chord steel-only truss structure (Scheme C)** | | |
| Chord | Outer diameter of chord tube $d_c$ | 12.68 (m) |
| | Wall thickness of chord tube $t_c$ | 91 (mm) |
| | Length of the tunnel $L_s$ | 1000 (m) |
| | Distance between upper chords $B$ | 80 (m) |
| | Section height $H$ | 100 (m) |
| Brace | Inner diameter of sloping brace tube $d_s$ | 11.00 (m) |
| | Wall thickness of sloping brace tube $t_s$ | 60 (mm) |
| | Inner diameter of horizontal brace tube $d_h$ | 11.00 (m) |
| | Wall thickness of horizontal brace tube $t_h$ | 60 (mm) |
| | Distance interval for each horizontal brace tubes $L_i$ | 100 (m) |
| **Triple-chord trussed CFDST hybrid structure (Scheme D)** | | |
| Chord | Outer diameter of outer tube $d_o$ | 13.66 (m) |
| | Inner diameter of inner tube $d_i$ | 12.50 (m) |
| | Wall thickness of outer tube $t_o$ | 40 (mm) |
| | Wall thickness of inner tube $t_i$ | 40 (mm) |
| | Wall thickness of sandwiched concrete $t_c$ | 500 (mm) |
| | Length of the tunnel $L_s$ | 1000 (m) |
| | Distance between upper chords $B$ | 80 (m) |
| | Section height $H$ | 20 (m) |
| Brace | Inner diameter of sloping brace tube $d_s$ | 11.00 (m) |
| | Wall thickness of sloping brace tube $t_s$ | 60 (mm) |
| | Inner diameter of horizontal brace tube $d_h$ | 11.00 (m) |
| | Wall thickness of horizontal brace tube $t_h$ | 60 (mm) |
| | Distance interval for each horizontal brace tubes $L_i$ | 100 (m) |

The wall thickness of the inner tube, outer tube and sandwiched concrete of Scheme B are determined by keeping the hollow ratio $\chi = (d_i + 2t_i)/(d_i + 2t_i + 2t_c)$ consistent with that of one chord tube of Scheme D.
*CFDST* concrete-filled double-skin tubular, *SFT* submerged floating tunnel.

therefore, we only need to investigate the temperature distribution in one cross-section. The temperature field distribution is acquired after 1.5-h of internal heating and the temperature-time curve is extracted from 3 typical detection points on the inner, middle and outer surfaces through the wall thickness.

## Calculation of parameter sensitivity value

The simplified Scheme D model served as the baseline for this analysis, with its design parameter $p$, structural material cost per unit length $c_s/L_s$ and maximum lateral displacement over the structural length $u_m/L_s$ used as reference values $p_0$, $c_{s0}/L_{s0}$ and $u_{m0}/L_{s0}$, respectively. We investigated the sensitivity to 13 parameters in total, as summarised in Fig. 5. For each parameter, five different models were established: one base model and four models with varied parameter values (two with increased values and two with decreased values). Apart from the parameter under analysis, all other design parameters remained consistent with the base model. The sensitivity value for each parameter is determined and compared using the following equation:

$$S_i = \frac{Y(X_0 + \Delta X_i) - Y(X_0)}{Y(X_0)} \Big/ \frac{\Delta X_i}{X_0} \tag{7}$$

where $X_0$ represents the baseline input value and $\Delta X_i$ is a perturbation in the baseline value $X_0$; $S_i$ is the sensitivity value for each perturbation of $\Delta X_i$. In this study, $Y$ is the material cost of the structure $c_s$ or the model prediction of $u_m$; $X$ is the value of each parameter $p$; $\Delta X_i$ is the perturbation in each parameter value $\Delta p_i$. For each parameter, we designed four models with altered parameter values. The average sensitivity value of these four models is used to represent the parameter sensitivity.

## Data availability

Test data used in this study are provided at https://doi.org/10.1016/j.istruc.2021.09.006 and https://doi.org/10.1016/j.jcsr.2015.03.002 respectively; the numerical simulation results can be found in supplementary information file; other data will be made available on request. The relevant request should be addressed to Fa-Cheng Wang (wangfacheng@tsinghua.edu.cn).

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

## Acknowledgements

This research was supported by the National Natural Science Foundation of China (No. 52425109 and No. 52371278). The financial support is highly appreciated.

## Author contributions

F.W. supervised the research process, discussed the results and revised the manuscript. T.Z. developed simplified fibre modelling method, validated numerical models, performed data analysis, carried out drawing-related tasks and wrote the manuscript. Z.C. assisted in the establishment of numerical models, discussed the results and revised the manuscript. L.H. assisted in the establishment of numerical models. J.Z. discussed results and revised the manuscript. L.G. discussed results and revised the manuscript.

## Competing interests

The authors declare no competing interests.
