## [Transparent Peer Review file · Communications Engineering]

Triple-chord trussed submerged floating tunnels: hybrid construction concept, feasibility and design

Corresponding Author: Dr Fa-Cheng Wang

Version 0:

Reviewer comments:

Reviewer #1

(Remarks to the Author)

1. Assumption of Equal Material Cost Across Schemes

The manuscript assumes that all four SFT schemes (Schemes A–D) are compared under the same material cost and traffic flow capacity. However, this assumption is not sufficiently justified. The structural configurations differ significantly — for example, Scheme D uses a triple-chord trussed CFDST system, which includes additional materials (e.g., two steel tubes, concrete infill, more joints) compared to the single-tube alternatives. Please clarify:

How was the “equal material cost” established?

Does this include differences in concrete and steel quantities, fabrication effort, and connection detailing?

If cost is assumed equivalent based on material unit price only, the limitations of this approach should be acknowledged.

2. Direct Comparison of Lateral Displacement and Stiffness Across Schemes

The direct comparison of lateral displacement and stiffness among Schemes A–D (Fig. 2c–d) lacks normalisation. Since Scheme D inherently has greater stiffness due to its trussed configuration and CFDST members, its lower displacement is expected. For a fairer evaluation of performance:

Consider normalising displacement using a reference value such as yield displacement, or comparing stress/strain utilisation ratios.

Alternatively, present displacement-to-cost or stiffness-to-weight efficiency indices to highlight material efficiency. This would help avoid conclusions that are a by-product of structural layout rather than design innovation.

3. Lack of Interaction Effects in Parameter Sensitivity Study

The parameter sensitivity analysis (Fig. 5) varies one parameter at a time, holding all others constant. However, for a hybrid structural system like the proposed SFT, performance arises from interdependent interactions among multiple parameters. The current OFAT (One-Factor-at-a-Time) method cannot capture these interactions. Please:

Clarify this limitation in the text.

Consider adding a note that future work could incorporate design of experiments (DOE) or multi-variable optimisation to reveal interaction effects more accurately.

4. Disjointed Discussion Section

The current Discussion section (Lines 326–361) largely repeats or extends the results rather than offering a broader interpretation. For better structure and clarity:

Separate results discussion from reflective analysis.

Use the Discussion to highlight implications, compare with literature, identify limitations (e.g., modelling simplifications), and suggest future work.

You may consider relocating the detailed parameter-based explanations back to the Results section to maintain thematic focus.

Reviewer #2

(Remarks to the Author)

The manuscript is well written and presents a novel concept using a double-skin concrete-filled steel tubular structure for submerged floating tunnels. This is a viable and innovative solution with potential advantages in structural performance. The methodology, along with the numerical and analytical assumptions used, are appropriate and commonly adopted in structural engineering practice. The following minor comments are recommended to further strengthen the manuscript:

1. The manuscript highlights three key performance aspects—lateral displacement, fire resistance, and fatigue—as the main advantages of the proposed concept. Could the authors elaborate on the rationale behind selecting these specific parameters? Are they considered the most critical for submerged floating tunnels based on previous studies, design codes, or practical considerations?
2. Please clarify whether the model is representative of a full-scale application or a scaled version, and if the latter, how scale effects might influence the results and their interpretation.
3. While the advantages of the double-skin system compared to conventional concrete or steel solutions are well illustrated, the manuscript would benefit from a discussion of the potential challenges or limitations associated with the proposed design. This could include aspects such as constructability, long-term durability, cost, or any unique installation issues expected in underwater environments.
4. The methods section could be improved by aligning it more clearly with the order of the results presented later in the manuscript. Currently, there appears to be some misalignment that could make it harder for readers to follow the workflow. Consider introducing a flowchart or schematic that outlines the overall methodology and links the applied methods with the corresponding outcomes.

Version 1:

Reviewer comments:

Reviewer #1

(Remarks to the Author)

The authors have satisfactorily addressed the concerns raised in the previous review. Key clarifications were provided regarding the material cost assumptions, comparative performance evaluation, and the limitations of the sensitivity analysis methodology. The revised manuscript is now clearer and better structured.

While the parametric study focuses primarily on single-parameter variations, the work still offers valuable initial insights into the behaviour of submerged floating tunnels (SFT) using hybrid CFDST trussed systems. This foundational understanding provides a good starting point for future research, including more advanced multi-variable optimisation and experimental validation. The methodology and findings will be informative to researchers and designers in the field of innovative long-span marine infrastructure, and help shape ongoing discussion around SFT feasibility.

I find the revised version suitable for publication and recommend acceptance.

Reviewer #2

(Remarks to the Author)

The authors have addressed the comments and the manuscript can be accepted for publication.

Thank you very much for taking the time to review our manuscript and for providing insightful and constructive comments. We greatly appreciate your efforts in helping us improve the quality and clarity of our work. Based on your suggestions, we have carefully revised the manuscript and addressed all the points you raised.

Please find below our detailed responses to your comments. All revisions made in the manuscript have been highlighted in yellow for your convenience.

Reviewer #1:

1. Comment: Assumption of Equal Material Cost Across Schemes

The manuscript assumes that all four SFT schemes (Schemes A–D) are compared under the same material cost and traffic flow capacity. However, this assumption is not sufficiently justified. The structural configurations differ significantly — for example, Scheme D uses a triple-chord trussed CFDST system, which includes additional materials (e.g., two steel tubes, concrete infill, more joints) compared to the single-tube alternatives. Please clarify:

How was the “equal material cost” established?

Does this include differences in concrete and steel quantities, fabrication effort, and connection detailing?

If cost is assumed equivalent based on material unit price only, the limitations of this approach should be acknowledged

Author’s Response:

We appreciate the reviewer for highlighting the need for a more thorough justification of the “equal material cost” assumption used in comparing the four SFT schemes (Schemes A–D). In response to this comment, we have revised the manuscript to provide additional clarification regarding the establishment of this assumption and to acknowledge its limitations. Our modifications are summarized as follows:

1) Clarification of the Equal Material Cost Basis:

In the revised manuscript, we have explicitly stated that the material costs are calculated based on market survey data and equations (5) and (6). Specifically, using the unit prices provided by the Beijing Municipal Commission of Housing and Urban-rural Development (430 RMB/m³ for concrete and 3803 RMB/t for steel). For each scheme, the quantities of concrete and steel are computed such that the overall material expenditure remains equivalent while maintaining the same traffic flow capacity. The design parameters listed in Table 2 are derived by ensuring that each SFT scheme operates under this material cost equivalence criterion.

2) Scope of Cost Factors Considered:

It is now more clearly indicated that the “material cost” in our analysis is determined solely from the direct costs of concrete and steel based on their respective market unit prices. This approach does not incorporate secondary cost factors, such as differences in concrete and steel quantities due to additional structural elements (e.g., extra steel tubes or additional connections in Scheme D), fabrication labour, or detailed connection processing costs. We have provided a rationale for this choice by noting that the evaluation primarily focuses on material cost discrepancies and their impact on structural performance (e.g., under lateral flow, vibration, and fire scenarios). A comprehensive economic analysis that includes construction and installation costs would require further investigation and is beyond the scope of the current study.

3) Acknowledgment of Limitations and Future Work:

We have expanded the discussion section to explicitly acknowledge that assuming cost equivalence based solely on material unit prices is a simplification. While this assumption allows for an initial comparative assessment, it does not capture the full complexity of cost influences arising from fabrication, assembly, and connection detailing. The revised discussion now highlights that although the present analysis provides valuable insights into material efficiency and performance trade-offs, future work will address a more detailed economic evaluation that includes construction processes, installation complexities, and the cost implications of connection detailing.

Revised Manuscript Excerpts

In the Results Section (Lines 218-228):

To ensure a fair comparison among the SFT schemes, the study assumes equal traffic flow bearing capacities and an equal material cost for each scheme, as shown in Fig. 2(a). Specifically, each scheme is designed to withstand a 6-lane traffic flow and the quantities of concrete and steel for each design are determined based on a market price investigation reported by the Beijing Municipal Commission of Housing and Urban-rural Development in September 2024⁵⁴, where the price of concrete is 430 RMB/m³ (60.38 USD/m³) and the price of steel is 3803 RMB/t (534.02 USD/t). This design criterion ensures that the overall material expenditure remains equivalent across the different schemes. It should be noted that this cost calculation considers only the raw material unit prices and does not incorporate additional factors such as fabrication effort, connection detailing or installation-related costs. The geometric details of the four schemes are listed in the Methods section. Additionally, we conduct a parametric sensitivity analysis on the maximum lateral displacement and cost under the lateral flow scenario to evaluate the adaptability and cost-effectiveness of the designs.

In the Discussion Section (Lines 416-420):

Several limitations of this study must also be acknowledged. First, while the simplified fibre modelling method offers computational efficiency for global analysis, it cannot capture local stress/strain distributions with sufficient precision, which would require full 3-D finite element modelling. Second, the parametric sensitivity analysis is conducted in an OFAT (One-Factor-at-a-Time) manner, which may neglect potential interaction effects among parameters. Future work should adopt design-of-experiments (DOE) or multi-variable optimisation techniques to provide a more holistic understanding of the parameter space. Third, the present comparison assumes equal material costs based on unit prices, without considering the differences in construction complexity, fabrication methods or connection detailing, which may influence total costs in practice. A more comprehensive economic analysis incorporating these factors is recommended to be pursued in future work.

In the Methods Section (Lines 585-594):

Assuming the SFT schemes can withstand a 6-lane traffic flow and share the same material cost, we designed four SFT schemes based on the Chinese standard JTJ 3370.1-2018⁶⁸ and the material price investigation reported by the Beijing Municipal Commission of Housing and Urban-rural Development in September 2024⁵⁴. Specifically, we calculate the material costs with the following equations:

For single-tube schemes:

$$\begin{cases} P_s = \frac{\pi}{4} \gamma_s \rho_s L_s [(d_i + 2t_i)^2 + (d_i + 2t_i + 2t_c + 2t_o)^2 - d_i^2 - (d_i + 2t_i + 2t_c)^2] \\ P_c = \frac{\pi}{4} \gamma_c L_s [(d_i + 2t_i + 2t_c)^2 - (d_i + 2t_i)^2] \\ P = P_s + P_c \end{cases} \quad (5)$$

For triple-chord trussed schemes:

$$\begin{cases} P_s = \frac{3\pi}{4} \gamma_s \rho_s L_s [(d_i + 2t_i)^2 + (d_i + 2t_i + 2t_c + 2t_o)^2 - d_i^2 - (d_i + 2t_i + 2t_c)^2] \\ P_c = \frac{3\pi}{4} \gamma_c L_s [(d_i + 2t_i + 2t_c)^2 - (d_i + 2t_i)^2] \\ P_b = \frac{\pi}{4} \gamma_s \rho_s \{n_s L_{sb} [(d_s + 2t_s)^2 - d_s^2] + n_h B [(d_h + 2t_h)^2 - d_h^2]\} \\ L_{sb} = \sqrt{(L_i/2)^2 + (B/2)^2 + H^2} \\ P = P_s + P_c + P_b \end{cases} \quad (6)$$

where, P represents the material cost of the scheme; P_s , P_c and P_b denote the material price of the steel used in chord tubes, the concrete used in chord tubes and the steel used in the braces respectively; γ_s and γ_c represent the unit price of steel and concrete respectively; n_s and n_h denote the number of sloping braces and horizontal braces used in the truss structures respectively; L_{sb} is the length of a single sloping brace.

We believe that these modifications address the reviewer's concerns and enhance the rigor and transparency of our analysis.

2. Comment: Direct Comparison of Lateral Displacement and Stiffness Across Schemes

The direct comparison of lateral displacement and stiffness among Schemes A–D (Fig. 2c–d) lacks normalisation. Since Scheme D inherently has greater stiffness due to its trussed configuration and CFDST members, its lower displacement is expected. For a fairer evaluation of performance:

Consider normalising displacement using a reference value such as yield displacement, or comparing stress/strain utilisation ratios.

Alternatively, present displacement-to-cost or stiffness-to-weight efficiency indices to highlight material efficiency. This would help avoid conclusions that are a by-product of structural layout rather than design innovation.

Author’s Response:

We appreciate the reviewer’s valuable feedback and would like to address the concerns regarding the comparison of lateral displacement and stiffness among the four SFT schemes. We would like to clarify two key points in response:

1) Elastic Displacement Response vs. Stress/Strain Capture:

We acknowledge that the methodology employed in this study primarily focuses on capturing the elastic displacement response of the SFT structures. Due to the simplifications inherent in this approach, stress and strain distributions are not accurately captured. Therefore, the yield displacement or stress/strain utilisation ratios can’t be acquired in this analysis. We have updated the manuscript, particularly in the methodology section, to explicitly clarify this limitation. While the current approach is focused on displacement, which is suitable for evaluating global structural performance, it does not provide a detailed representation of stress and strain behaviours, which would require a more complex modelling approach.

2) Justification for Direct Comparison of Displacement and Stiffness:

We have ensured that the material costs are identical across all four SFT schemes, as outlined in the manuscript. Given this condition, the direct comparison of

displacement and stiffness is valid. In essence, comparing displacement or stiffness values directly is equivalent to using displacement-to-cost or stiffness-to-cost ratios, as both approaches would yield the same conclusions when the material costs are equal. Therefore, there is no need to introduce additional normalisation or cost-performance indices in this study, as the primary differences in displacement and stiffness are due to the design innovations rather than material usage (cost). We have updated the manuscript to more clearly explain this rationale, ensuring transparency in our comparison method.

In light of these, we have made the following revisions to the manuscript to address the reviewer's concerns:

1) Methodology Clarification:

We have added a clear statement on methodology introduction to acknowledge that the model is focused on capturing the elastic displacement response, and that stress and strain behaviours are not accurately modelled in this study due to the simplifications inherent in the approach.

2) Justification for Direct Comparison of Displacement and Stiffness:

We have rephrased the conclusion to emphasise that, given the equal material costs, comparing displacement and stiffness directly is equivalent to comparing the displacement-to-cost or stiffness-to-cost ratios. Therefore, the direct comparison is valid and does not require additional normalisation.

Revised Manuscript Excerpts

*In the **Results** Section (Lines 201-205):*

The validity of the simplified fibre modelling method is demonstrated through comparisons with the 3-D FE modelling method; the details are provided in the Methods Section. It should be noted that this simplified modelling method primarily focuses on capturing the elastic displacement response of the SFT structures. However, the assumptions made in this simplification introduce limitations in accurately capturing detailed stress and strain distributions, which would require more advanced modelling methods such as 3-D FE modelling method.

In the Results Section (Lines 248-256):

Fig. 2(c) and Fig. 2(d) show the comparison of the maximum lateral displacement and the structural stiffness for all four schemes. This comparison is made under the condition that material costs and traffic flow bearing capacities are the same for all schemes. As such, the direct comparison of displacement and stiffness not only reflects the structural performance but also indirectly indicates the material efficiency of each design. Therefore, while normalising these parameters or comparing them against cost metrics is a valid approach in certain contexts, the direct comparison remains appropriate and informative for evaluating the designs in this study, as material cost equivalence ensures that any differences observed are due to the structural configurations themselves.

3. Comment: Lack of Interaction Effects in Parameter Sensitivity Study

The parameter sensitivity analysis (Fig. 5) varies one parameter at a time, holding all others constant. However, for a hybrid structural system like the proposed SFT, performance arises from interdependent interactions among multiple parameters. The current OFAT (One-Factor-at-a-Time) method cannot capture these interactions. Please: Clarify this limitation in the text.

Consider adding a note that future work could incorporate design of experiments (DOE) or multi-variable optimisation to reveal interaction effects more accurately.

Author's Response:

We appreciate the reviewer's constructive feedback on the parameter sensitivity analysis. We acknowledge that the current One-Factor-at-a-Time (OFAT) method, which varies one parameter at a time while holding all others constant, may not fully capture the complex interdependencies among multiple parameters, especially in hybrid structural systems like the proposed SFT. This limitation is recognised in the study, and we agree that incorporating more advanced techniques could offer a better understanding of these interactions.

In response to the reviewer's suggestion, we have clarified this limitation in the revised manuscript. Specifically, we have made the following revisions to the manuscript:

1) Clarification of the OFAT Method Limitation:

We have updated the discussion to include a clear statement regarding the limitations of the OFAT method in capturing parameter interactions and noted that this is a simplification of the real system behaviour.

2) Future Work Suggestions:

We have suggested that future research could involve DOE or multi-variable optimisation to more accurately capture the interactions between multiple design parameters and refine the optimisation of the SFT structure.

Revised Manuscript Excerpt

In the Discussion Section (Lines 412-416):

Second, the parametric sensitivity analysis is conducted in an OFAT (One-Factor-at-a-Time) manner, which may neglect potential interaction effects among parameters. Future work should adopt design-of-experiments (DOE) or multi-variable optimisation techniques to provide a more holistic understanding of the parameter space.

4. Comment: Disjointed Discussion Section

The current Discussion section (Lines 326–361) largely repeats or extends the results rather than offering a broader interpretation. For better structure and clarity:

Separate results discussion from reflective analysis.

Use the Discussion to highlight implications, compare with literature, identify limitations (e.g., modelling simplifications), and suggest future work.

You may consider relocating the detailed parameter-based explanations back to the Results section to maintain thematic focus.

Author's Response:

We thank the reviewer for the insightful suggestion. We fully agree that the current Discussion section overly reiterates the results and would benefit from a clearer separation between results interpretation and reflective analysis. In response, we have substantially revised the manuscript to enhance its structure and clarity:

1) Removed the parameter-based explanations to Results section:

The detailed parameter-based explanations—originally embedded within the Discussion section—has now been relocated to the Results section, where it better aligns with the presentation of simulation outcomes (especially in relation to Fig. 5).

2) Rewrote the Discussion section:

The Discussion section has been revised to focus more on implications, theoretical insights, comparisons with existing literature, and a critical reflection on limitations and directions for future research.

Revised Manuscript Excerpt

*In the **Results Section (Lines 358-369):***

The results reveal that the ratio of maximum lateral displacement to the structural length u_m/L_s is positively related to L_s , H and L_i , and negatively related to the remaining 10 parameters, while the material cost to structural length c_s/L_s ratio is negatively related to L_i and positively related to the remaining 10 parameters. Among all the parameters mentioned above, **three parameters—the length of the tunnel L_s , section**

height H , and distance between upper chords B —exert the most significant influence on the maximum lateral displacement, while introducing negligible increases in material cost. These parameters are therefore recommended as primary variables for displacement control in practical design optimisation. Specifically, increasing L_s and H leads to higher displacement, whereas increasing B can mitigate this effect. Additionally, benefiting from the relative lower price of concrete, increasing the thickness of the sandwiched concrete t_c causes a limited cost rise but brings about a relatively notable increase in stiffness. Accordingly, t_c may also be considered a favourable design parameter for improving lateral displacement performance. For the remaining nine parameters, given their low sensitivity and limited cost-benefit potential, design adjustments are not recommended for displacement control.

In the Discussion Section (Lines 379-440):

This study demonstrates that the proposed triple-chord trussed CFDST hybrid SFT design offers considerable structural performance advantages under typical and potential service scenarios, including lateral flow, vibration and internal fire, while maintaining material cost parity with existing solutions. These performance gains are primarily attributed to the synergistic action of the steel and concrete and to the truss-based configuration of the system.

From a structural perspective, the proposed design significantly enhances lateral stiffness and reduces deformation without excessive material usage. In particular, its response under lateral flow benefits from the flexibility of the truss layout and the efficient use of sandwiched concrete, which outperforms both single-tube and steel-only truss schemes. Furthermore, under cyclic loading, the integration of concrete in the joints mitigates stress concentrations and prolongs fatigue life—a conclusion consistent with previous findings on concrete-filled tubular joints^{10,11,26}. In fire scenarios, the low thermal conductivity of the concrete acts as a natural insulator, protecting the outer tubes which are critical for structural strength. This inherent fire resistance may allow designers to avoid expensive protective coatings required for steel-only alternatives.

Although increasing attention has been given to the structural performance of submerged floating tunnels in recent years, most existing studies remain focused on single-tube configurations^{7-9,16-20}. Research on steel–concrete composite schemes is still limited¹⁵, and investigations of trussed hybrid systems are even rarer. However, single-tube solutions face significant challenges in harsh marine environments, and no SFT projects have yet been successfully realised in practice. Our design exemplifies the potential of hybrid composite systems in addressing the multifaceted demands of subsea infrastructure. The study provides a proof of concept for combining structural and economic efficiency through composite interaction and a rational geometric configuration, offering a promising design option for the practical application of SFT systems. Additionally, the use of a multi-scale numerical analysis methodology contributes a feasible framework for evaluating such complex systems that balances computational efficiency with modelling depth.

Several limitations of this study must also be acknowledged. First, while the simplified fibre modelling method offers computational efficiency for global analysis, it cannot capture local stress/strain distributions with sufficient precision, which would require full 3-D finite element modelling. Second, the parametric sensitivity analysis is conducted in an OFAT (One-Factor-at-a-Time) manner, which may neglect potential interaction effects among parameters. Future work should adopt design-of-experiments (DOE) or multi-variable optimisation techniques to provide a more holistic understanding of the parameter space. Third, the present comparison assumes equal material costs based on unit prices, without considering the differences in construction complexity, fabrication methods or connection detailing, which may influence total costs in practice. A more comprehensive economic analysis incorporating these factors is recommended to be pursued in future work. Fourth, while full-scale geometry is adopted for the lateral displacement, thermal, and parameter sensitivity analyses, a scaled-down model is used in the fatigue analysis. This is because accurate fatigue assessment requires capturing hotspot stresses, which simplified fibre models cannot provide, and full-scale fatigue simulations are extremely time-consuming. It is

acknowledged that fatigue performance generally degrades with increasing scale due to material imperfections and enlargement of high-stress zones. As such, the small-scale model may yield an overestimated fatigue life. However, since this part of the study focuses more on comparative trends across schemes, the adopted scale remains suitable for relative performance evaluation. A systematic scale-effect analysis—integrating theoretical modelling, physical testing and numerical simulation—would provide valuable insights and is recommended for future work. Finally, while the proposed design demonstrates better performance in the numerical evaluations, several implementation challenges merit attention. The composite double-skin configuration—particularly the use of CFDST tubes with truss bracing—may introduce complexities compared to steel-only design in prefabrication, construction and underwater assembly. Ensuring reliable concrete filling, joint integrity and corrosion protection under deep-sea conditions presents non-trivial engineering challenges. Additionally, long-term maintenance and durability issues, especially in aggressive marine environments, could influence lifecycle performance. These aspects lie beyond the scope of the current study but are critical for future experimental validation and engineering application.

We hope these changes can improve the thematic consistency of the manuscript and align with the standard practice in research reporting.

Thank you very much for taking the time to review our manuscript and for providing insightful and constructive comments. We greatly appreciate your efforts in helping us improve the quality and clarity of our work. Based on your suggestions, we have carefully revised the manuscript and addressed all the points you raised.

Please find below our detailed responses to your comments. All revisions made in the manuscript have been highlighted in yellow for your convenience.

Reviewer #2:

The manuscript is well written and presents a novel concept using a double-skin concrete-filled steel tubular structure for submerged floating tunnels. This is a viable and innovative solution with potential advantages in structural performance. The methodology, along with the numerical and analytical assumptions used, are appropriate and commonly adopted in structural engineering practice. The following minor comments are recommended to further strengthen the manuscript:

1. Comment: The manuscript highlights three key performance aspects—lateral displacement, fire resistance, and fatigue—as the main advantages of the proposed concept. Could the authors elaborate on the rationale behind selecting these specific parameters? Are they considered the most critical for submerged floating tunnels based on previous studies, design codes, or practical considerations?

Author's Response:

We thank the reviewer for raising this important point. We agree that clarification is needed regarding the rationale behind selecting lateral displacement, fatigue life, and fire resistance as the three main performance aspects in this study. In response, we have revised the Introduction to explicitly explain why these three performance criteria were chosen. Specifically:

1) Lateral displacement:

This is a critical serviceability requirement for SFT structures subjected to lateral

flow loads. Excessive deformation could compromise tunnel functionality and traffic safety. Displacement limits under lateral environmental loads are commonly prescribed in design standards for long-span transportation infrastructure.

2) Fatigue performance:

SFT structures are subject to continuous dynamic excitation from waves and internal traffic. These cyclic loads make fatigue, particularly in the joint regions of trussed systems, a key durability concern. This has been highlighted in multiple previous studies and is particularly relevant for the proposed design, which employs steel-concrete composite joints.

3) Fire resistance:

Due to the enclosed and isolated underwater environment of SFTs, internal fire scenarios present a severe safety hazard with limited escape routes and potential structural collapse. This is a critical consideration in tunnel design codes worldwide. Moreover, failure of an SFT during fire could result in catastrophic water ingress, making fire performance a uniquely important aspect of safety in submerged tunnel systems.

We have updated the Introduction section to include these justifications and to better link them to previous studies and practical design requirements.

Revised Manuscript Excerpt

In the Introduction Section (Lines 54-86):

However, SFT are also exposed to a unique combination of loading from both normal service and hazardous scenarios due to their distinct service environments, typically including:

- **High-velocity lateral flows in fjord-like terrains**, which can induce large lateral displacements that threaten both the operational safety and long-term serviceability of the tunnel. Excessive deformation under lateral flow may interfere with internal traffic or even compromise structural stability. Design guidelines for long-span transportation infrastructure typically impose strict displacement limits to mitigate such risks⁶. For SFTs directly exposed to

marine currents, lateral displacement thus represents a fundamental performance concern and should be treated as a primary serviceability criterion in SFT design.

- **Wave-induced vibrations**, which give rise to cumulative fatigue damage, especially in structural systems featuring multiple connections, such as trusses. Unlike immersed tunnels, which benefit from soil confinement, SFTs remain continuously subjected to dynamic excitation from waves^{1,5,7,8} and internal traffic⁹. Previous research has shown that fatigue cracking in joint regions often governs the service life of marine truss structures^{10,11}. Accordingly, fatigue life at critical connection locations is commonly selected as a key durability metric in the evaluation of SFT designs.
- **Internal fires occurring within the enclosed underwater space**, which pose severe threats to occupant safety and structural integrity. Compared to open-air bridges, SFTs have more confined ventilation conditions, and their direct contact with external water increases the consequence of structural failure due to thermal degradation. A breach of the tunnel wall during a fire could result in uncontrolled water ingress and rapid flooding. For this reason, fire resistance is commonly regarded as a critical requirement in tunnel design practice, especially for the SFT structures¹²⁻¹⁵.

These harsh loading conditions introduce heightened design complexity and operational challenges compared to conventional cross-sea structures. While SFT offer the potential for superior performance, addressing these challenges requires design innovation and thorough comparative analyses to ensure safety and feasibility. Meanwhile, although the mentioned scenarios—lateral flow, wave-induced vibration, and internal fire—do not represent the full spectrum of loads that an SFT may be subjected to, they are among the most representative and critical. Therefore, this study focuses on these three scenarios as the principal basis for performance evaluation.

2. Comment: Please clarify whether the model is representative of a full-scale application or a scaled version, and if the latter, how scale effects might influence the results and their interpretation.

Author's Response:

We thank the reviewer for raising this important point. We confirm that the numerical modelling of the lateral displacement response analysis under lateral flow and the thermal performance analysis under internal fire was conducted on a full-scale SFT model. The parametric sensitivity analysis under lateral flow was also performed on the full-scale configuration.

However, for the fatigue analysis under wave-induced vibrations, a scaled-down structural model was adopted. This decision was made for the following reasons:

1) Accuracy and computational feasibility:

Fatigue life prediction requires accurate stress resolution at joint hotspots. The simplified fibre-based modelling approach used for global analysis is not capable of capturing these local stress effects reliably. Employing a full 3-D finite element model at full scale for fatigue analysis would impose a prohibitive computational cost. We acknowledge this limitation and have added a corresponding clarification to the revised Discussion section.

2) Potential overestimation introduced by scaling effects:

In general, larger-scale joints typically exhibit lower fatigue resistance than their small-scale counterparts due to increased probability of internal material defects, enlargement of high-stress zones, and the cumulative effects of fabrication-related surface imperfections. Thus, the use of a small-scale model for fatigue analysis may overestimate absolute fatigue life. This point is now clarified in the revised manuscript. Nevertheless, it should also be noted that since our primary goal is to compare the relative fatigue performance across different structural schemes, the adopted approach still offers valid and informative insights.

3) Scope of scaling effect modelling:

A rigorous scale effect analysis requires detailed theoretical, numerical and experimental investigations, which are beyond the scope of the current study. We have acknowledged this as a limitation in the revised Discussion and suggested it as an important avenue for future work.

Revised Manuscript Excerpt

In the Discussion Section (Lines 420-431):

Fourth, while full-scale geometry is adopted for the lateral displacement, thermal, and parameter sensitivity analyses, a scaled-down model is used in the fatigue analysis. This is because accurate fatigue assessment requires capturing hotspot stresses, which simplified fibre models cannot provide, and full-scale fatigue simulations are extremely time-consuming. It is acknowledged that fatigue performance generally degrades with increasing scale due to material imperfections and enlargement of high-stress zones. As such, the small-scale model may yield an overestimated fatigue life. However, since this part of the study focuses more on comparative trends across schemes, the adopted scale remains suitable for relative performance evaluation. A systematic scale-effect analysis—integrating theoretical modelling, physical testing and numerical simulation—would provide valuable insights and is recommended for future work.

3. Comment: While the advantages of the double-skin system compared to conventional concrete or steel solutions are well illustrated, the manuscript would benefit from a discussion of the potential challenges or limitations associated with the proposed design. This could include aspects such as constructability, long-term durability, cost, or any unique installation issues expected in underwater environments.

Author's Response:

We appreciate the reviewer's thoughtful observation. We agree that, in addition to highlighting the performance advantages of the proposed hybrid system, it is also important to reflect on the potential implementation challenges. In response, we have expanded the contents in the Discussion section to address issues such as constructability, long-term durability, and installation in underwater environments. These considerations are relevant for practical application and align with the broader goal of assessing feasibility in real-world conditions. We hope this additional discussion improves the balance and completeness of the manuscript.

Revised Manuscript Excerpt

In the Discussion Section (Lines 431-440):

Finally, while the proposed design demonstrates better performance in the numerical evaluations, several implementation challenges merit attention. The composite double-skin configuration—particularly the use of CFDST tubes with truss bracing—may introduce complexities compared to steel-only design in prefabrication, construction and underwater assembly. Ensuring reliable concrete filling, joint integrity and corrosion protection under deep-sea conditions presents non-trivial engineering challenges. Additionally, long-term maintenance and durability issues, especially in aggressive marine environments, could influence lifecycle performance. These aspects lie beyond the scope of the current study but are critical for future experimental validation and engineering application.

4. Comment: The methods section could be improved by aligning it more clearly with the order of the results presented later in the manuscript. Currently, there appears to be some misalignment that could make it harder for readers to follow the workflow. Consider introducing a flowchart or schematic that outlines the overall methodology and links the applied methods with the corresponding outcomes.

Author's Response:

We thank the reviewer for this helpful suggestion. We agree that aligning the structure of the Methods section with the sequence of results enhances clarity and improves the overall coherence of the manuscript. In response, we have introduced a new figure (Fig. 6) that visually summarises the overall methodology adopted in the study. This schematic outlines the distinct loading scenarios (lateral flow, vibration, and internal fire), the corresponding numerical methods employed (3-D finite element modelling and simplified fibre modelling), and the specific structural behaviours analysed in each case. To complement this visual representation, we have added an introductory paragraph at the beginning of the Methods section to clearly explain the modelling logic and how each method connects to the relevant performance evaluations discussed in the results. These changes aim to provide readers with a clear, intuitive understanding of the analytical workflow used in this study.

Revised Manuscript Excerpt

*In the **Methods** Section (Lines 443-458):*

The methodology adopted in this study involves a multi-scale modelling framework that integrates different numerical approaches to evaluate the performance of the proposed hybrid SFT structure under distinct loading conditions. As summarised in Fig. 6, we focus on three critical service scenarios—lateral flow, wave-induced vibration, and internal fire—each of which utilises a separate to a simulation approach. Specifically, 3-D FE modelling is used to analyse the local structural behaviour, including fatigue life in joint regions and temperature distribution under fire. In parallel, a simplified fibre modelling method is employed to evaluate the global elastic response of the full SFT structure, particularly the lateral displacement under hydrodynamic

loading.

Fig. 6 | Summary of the methodology of the research. Unique loading combinations of lateral flow, vibration and internal fire are investigated. We use 3-D FE modelling to simulate the local behaviour of the structure, including the fatigue life of the joint regions and the temperature distribution under internal fire loading, and simplified fibre modelling to simulate the global behaviour, including determining the maximum lateral displacement.

Reviewer #1:

The authors have satisfactorily addressed the concerns raised in the previous review. Key clarifications were provided regarding the material cost assumptions, comparative performance evaluation, and the limitations of the sensitivity analysis methodology. The revised manuscript is now clearer and better structured.

While the parametric study focuses primarily on single-parameter variations, the work still offers valuable initial insights into the behaviour of submerged floating tunnels (SFT) using hybrid CFDST trussed systems. This foundational understanding provides a good starting point for future research, including more advanced multi-variable optimisation and experimental validation. The methodology and findings will be informative to researchers and designers in the field of innovative long-span marine infrastructure, and help shape ongoing discussion around SFT feasibility.

I find the revised version suitable for publication and recommend acceptance.

Author's Response:

We thank for your thoughtful and encouraging comments. We sincerely appreciate the time and effort you have devoted to reviewing our work, and are grateful for your recognition of its potential value to the field.

Reviewer #2:

The authors have addressed the comments and the manuscript can be accepted for publication.

Author's Response:

We thank for your time and consideration in reviewing our manuscript. We appreciate your positive assessment and are glad that our work has met your expectations.